# Accuracy of Dose Administered to Children Using Off-Labelled or Unlicensed Oral Dosage Forms

**DOI:** 10.3390/pharmaceutics13071014

**Published:** 2021-07-02

**Authors:** Guillaume Binson, Cécile Sanchez, Karen Waton, Adeline Chanat, Massimo Di Maio, Karine Beuzit, Antoine Dupuis

**Affiliations:** 1Pharmacy, Poitiers Teaching Hospital, 86000 Poitiers, France; cecile.sanchez@chu-poitiers.fr (C.S.); karen.waton@chu-poitiers.fr (K.W.); adeline.chanat@chu-poitiers.fr (A.C.); karine.beuzit@chu-poitiers.fr (K.B.); antoine.dupuis@chu-poitiers.fr (A.D.); 2School of Medicine and Pharmacy, University of Poitiers, 86000 Poitiers, France; 3Pediatrics Department, Nîmes Teaching Hospital, 30000 Nîmes, France; massimo.dimaio@chu-nimes.fr

**Keywords:** administration management, drug administration, pediatric medicine, personalized medicine, off-labelled dosage forms

## Abstract

The pediatric population suffers from a lack of age-appropriate medicines leading to unsafe situations when off-labelled or unlicensed drugs are used. Assessing the best option to administrate medicines when manipulations are required is essential in order to improve child care. This study aimed to compare the accuracy of the administered dose provided by three dosage forms and their techniques of administration. Different techniques of administration were assessed, covering three oral dosage forms (commercially available tablets, capsules, oral suspensions) using two APIs not available in a children-adapted dosage form. Techniques of administration were simulated and administered doses were determined using HPLC-UV. Means were compared to the target dose while distributions of doses were compared between each technique. For both APIs, mean administered doses obtained with capsules and tablets were significantly different from the target dose, whereas there was no statistical difference with oral suspensions. Distributions of doses showed significant difference between the three dosage forms. This study demonstrates that manipulations of solid oral dosage forms provide dramatic underdosing leading to unsafe situations. Compounded oral suspension is the best option to avoid underdosing and dose variation. This solution should be prioritized when age-appropriate commercial medicines are not available.

## 1. Introduction

The pediatric population, especially young infants such as neonates, presents a particular challenge when administration of an oral medicine is required. Several factors such as the physiological variability of this population or specific factors may hamper the administration of oral medications [1,2,3]. For instance, most children under the age of six are unable to swallow oral solid forms even if specific training is provided [4]. In the past few years, several developments have been made to improve the acceptability of medicines to children, such as mini-tablets [5], but most commercially available medications remain designed for the adult population in terms of dose and galenic forms, complicating the management of pediatric disease by a lack of age-appropriate formulations for children [6]. This issue leads clinicians to prescribe off-labeled or unlicensed (OLUL) medicines with limited evidence on safety, accuracy and efficacy, and forces caregivers or parents to manipulate drugs so as to render them administrable [7,8,9]. The term «manipulation» may be defined by a physical alteration of a drug dosage form aimed to obtain the required dose and/or to facilitate oral administration when using an unsuitable dosage form. Consequently to the lack of age-appropriate dosage forms, tablets represent the most frequently used dosage form in pediatric practices but are used with manipulations such as cutting, dispersing or crushing [10]. This common practice leads to unevaluated situations with no evidence about the accuracy of the administered dose. Several studies highlighted the difficulty to obtain an accurate dose from manipulated tablets with large dose deviations and discrepancies between manipulations due to the galenic, the prescribed dose or the experience of the manipulator [11,12,13]. Other oral dosage forms may encounter the same problem. For instance, a capsule may require manipulations to be administered or to obtain a smaller dose from a commercially available medication. In that case, strength adjustment may be obtained by opening the capsules and dividing the powder [14]. Here, again, this leads to an unevaluated situation questioning the security of these handlings.

Given this lack of age-appropriate strengths and dosage forms, several tools for personalized medicine in pediatrics could be considered, such as minitablets or 3D-printing medicines [15,16], but pharmaceutical compounding represents a more commonly used alternative. This approach has the advantage of easily providing adequate strength and dosage form which can often be used without manipulations [9]. Nevertheless, pharmaceutical compounding does not resolve all issues. Indeed, capsules are one of the most compounded dosage form for the pediatric population, leading to the same administration issues as that described above [17]. In that case, the only advantage consists of directly possessing the adequate dose of API to dilute into food or beverage before administration. Finally, compounded liquid oral dosage forms such as syrups or suspensions facilitate dose fractioning and are less associated with administration issues related to pediatric physiological limits.

While the use of OLUL medicine is common in the pediatric population [18], it has been associated as a risk factor for adverse drug event (ADE), especially due to the use of inappropriate dosage forms [19]. Hence, it is mandatory to assess if manipulations of oral dosage forms may be associated with dose issues leading to potential ADE and/or treatment inefficacy. Therefore, the aim of our study was first to assess the accuracy of the administered dose provided by manipulations of two oral solid dosage forms from OLUL medicine commonly used in young children. Then, we aimed to propose an alternative, using compounded oral suspensions, to reduce dose inaccuracy associated with the use of OLUL medicine.

## 2. Materials and Methods

To mimic the different administration techniques used in pediatric wards, we chose to focus on two APIs: spironolactone and hydrochlorothiazide. Indeed, these APIs are commonly used in neonates [20] and none are commercially available under a formulation suitable for the pediatric population. We chose to mimic the administration of 6.25 mg for both drugs, using the 25 mg scored tablets commercially available as well as compounded capsules and oral suspensions. This dose was chosen according to the range of doses required for neonates (1–4 mg/kg/day for spironolactone, 2–4 mg/kg/day for hydrochlorothiazide) and since it is the smallest achievable dose from one quarter of commercialized tablets [20,21].

### 2.1. Study Design

#### 2.1.1. Dosage Forms

*Commercially available oral solid forms*: Hydrochlorothiazide (Esidrex^®^) 25 mg and spironolactone (Aldactone^®^) 25 mg tablets were obtained from Juvise Pharmaceuticals (Villeurbanne, France) and Pfizer (Paris, France), respectively.

*Compounded capsules*: For each API, three batches of 100 capsules at 6.25 mg were compounded according to French good practices of compounding [22]. Capsules were compounded from API raw powder with lactose as excipient by empowered staff. Uniformity of mass and uniformity of content assays were performed on each batch according to the recommendations of the European Pharmacopeia [23,24].

*Compounded oral suspensions*: As for capsules, three batches of oral suspension were compounded for each API. Targeted API concentrations were 2 and 5 mg/mL for hydrochlorothiazide and spironolactone, respectively. Suspensions were compounded from API raw powder and Syrspend^®^ SF PH4 Dry as suspending agent. Shelf life of both suspensions were assessed for 60 days [25]. From each batch, three samples were collected to assess strength and homogeneity using HPLC-UV analysis. Acceptance criteria was set as 90–110% of the theoretical value for strength and homogeneity was achieved if the coefficient of variation between samples was lower than 10% [26]. The same criteria were applied for capsules assays.

#### 2.1.2. Mimicking Administration Modalities

*Tablets*: According to the literature [27,28], two methods were applied to simulate the administration of tablets to neonates. The first consisted of crushing the whole tablets with a specific crushing device (Pilldrink^®^, Inresa, Bartenheim, France). The obtained powder was mixed with 4 mL of drinking water and a fraction of the obtained suspension was collected with a 1 mL enteral syringe (Asept InMed, Quint-Fonsegrives, France). The volume was then put into a volumetric flask to mimic child administration. The second method consisted of quartering tablets with a cutting device (Merk, Semoy, France) and crushing them separately with the crushing device used above. Again, the obtained powder was mixed with drinking water and the obtained suspension was put into a volumetric flask with a 1 mL enteral syringe to mimic administration and to allow API dilution. For each method, 30 tablets were assessed (*n* = 30).

*Capsules*: 10 capsules from each of the three batch were individually assessed (*n* = 30). According to the literature [28,29], capsule administration was mimicked as follows: a capsule was manually emptied by opening the shell over a plastic cup to recover the maximum amount of API; then, 1 mL of drinking water was added and mixed with the recovered powder. The maximum volume was then collected using a 1 mL syringe for enteral administration and put into a volumetric flask to mimic child administration. For each API, 30 capsules were used in a simulation of administration.

*Oral suspensions*: The volume required to obtain the target dose of API (6.25 mg) was collected using the adequate enteral syringe (Asept InMed, Quint-Fonsegrives, France) connected to the container through a syringe/bottle adapter (Vygon, Ecouen, France) after vigorous manual agitation to ensure APIs resuspension. Collected volumes were then put into volumetric flasks. For each batch, 10 samplings were performed (*n* = 30).

For all administration modalities, samples were withdrawn immediately after mixing. No investigation into time to sample versus dose recovered were performed. Administration modalities are summarized in Figure 1.

#### 2.1.3. Drug Content Determination

After simulations of administration, samples were collected inside a volumetric flask and then diluted. Spironolactone samples were diluted with methanol (1:250 ratio) while hydrochlorothiazide samples were diluted with H_2_O:methanol 90:10 (1:125 ratio). Samples were then centrifugated (4000× *g*) for 1 min and supernatants were injected onto the HPLC column to determine API content. This determination was assessed by previously validated analytical methods [25]. Briefly, HPLC-UV analyses were performed using a chromatographic system and separation was provided by a Purospher^®^ STAR RP-10 endcapped (5 µm) 150 × 4.6 mm column. Detection wavelengths were set at 224 nm for hydrochlorothiazide and 238 nm for spironolactone. The mobile phase was composed of methanol and water (70:30, *v*:*v*) for spironolactone and methanol and water (20:80, *v*:*v*) with pH adjusted to 4.5 with acetic acid for hydrochlorothiazide. These methods were fully validated according to international guidelines [30].

### 2.2. Statistical Analysis

Statistical analysis was performed using R software. Mean administered doses were compared to the theoretical value (6.25 mg) using a *Z*-test (α = 5%). Distributions of doses were compared two by two using a Mann–Whitney test and, globally, a Kruskal–Wallis test (α = 5%). Average contents obtained after administration simulations were compared to target value (6.25 mg) using a Student *t*-test (α = 5%).

## 3. Results

### 3.1. Quality of Spironolactone and Hydrochlorothiazide Compounded Forms

*Capsules*: For capsules, average APIs strengths are summarized in Table 1 with a targeted strength of 6.25 mg for both hydrochlorothiazide and spironolactone. Whether for uniformity of content assay or uniformity of mass assay, no capsules were found outside the limits set by the European Pharmacopeia.

*Oral suspensions*: Average API contents for oral suspensions are summarized in Table 2. Targeted API concentration was set at 2 mg/mL for hydrochlorothiazide and 5 mg/mL for spironolactone. No batches were found outside the 10% range around the theoretical content and coefficient of variations remained under 10%.

### 3.2. Accuracy of Administered Doses

*Spironolactone*: After simulating administrations, the mean recovered dose was 4.62 ± 1.38 mg and 5.17 ± 1.52 mg with quartered and crushed tablets, respectively, 1.05 ± 0.48 mg with capsules and 6.11 ± 0.30 mg with oral suspensions. Mean recovered doses obtained with capsules, quartered tablets and crushed tablets were significantly different from the theoretical dose (*p* < 0.01), while mean recovered dose obtained with oral suspensions showed no significant difference (*p* = 0.70). Distributions of analyzed samples are shown in Figure 2. The narrowest distribution was observed with oral suspensions while the widest was observed with crushed tablets. The Kruskal–Wallis test showed a significant difference between the four administration modality dose distributions (*p* < 0.01). Mann–Whitney testing between in pair distributions likewise assessed significant differences (*p* < 0.01). The smallest difference was observed between quartered and crushed tablets (*p* = 0.009).

*Hydrochlorothiazide*: After simulating administrations, the mean recovered dose was 4.19 ± 0.96 mg and 5.47 ± 0.85 mg with quartered and crushed tablets, respectively, 3.05 ± 0.84 mg with capsules and 6.26 ± 0.35 mg with oral suspensions. Mean recovered doses obtained with capsules, quartered tablets and crushed tablets were significantly different from the theoretical dose (*p* < 0.01) while mean recovered dose obtained with oral suspensions showed no significant difference (*p* = 0.67). Distributions of analyzed samples are shown in Figure 3. The narrowest distribution was observed with oral suspensions while the widest was observed with crushed tablets. The Kruskal–Wallis test showed a significant difference between the four administration modality dose distributions (*p* < 0.01). Mann–Whitney testing between in pair distributions also assessed significant differences (*p* < 0.01) except between quartered and crushed tablets (*p* = 0.28).

## 4. Discussion

In 1999, the Institute for Safe Medication Practices issued a statement about the five rights of medication use: right patient, drug, time, dose and route [31]. They drew attention to the importance of using adequate practices to perform drug administration and avoid medication errors. Nevertheless, if these five rights are regarded as standards for safe medication practices, they focus only on individual performance and fail to raise concerns regarding processes and organization. In fact, providing the right dose to the right patient is not just a matter choosing the right dosage of drug. It also implies drug reconstitution, dilution and occasional manipulations of drug dosage forms, especially with the pediatric population [10]. In this case, the concept of ‘right dose’ could be defined by giving a precise and true dose to the patient where precision represents the variability of values and trueness the adequacy between expected and measured values. A drug manipulation performed several times would always give the same expected dose.

In this study, four practices of administration, including drug manipulations, were assessed covering most of the dosage forms potentially used in the pediatric population [27]. These assays were performed on two different APIs, spironolactone and hydrochlorothiazide, commonly used in the pediatric population, especially in neonates [20]. They are not available with a pediatric-adapted medication, in terms of strength and dosage forms, leaving caregivers to manipulate already commercialized drugs (crushing or quartering tablets) or to ask for pharmaceutical compounding of these APIs (oral suspensions or capsules). Considering these different options, our results clearly demonstrate that the administration of oral suspensions is more likely to obtain the required dose, with the best precision and trueness.

The discrepancies between these different dosage forms could be explained by several factors. First, spironolactone and hydrochlorothiazide are hydrophobic compounds displaying poor to very poor solubility in water [32]. This property is involved in the formation or precipitates and agglomerates when mixing these APIs with water, leading to the formation of an inhomogeneous suspension. This prevents to obtain adequate reproducible doses and the formation of precipitates prevents nurses and caregivers from fully recover the target dose. On the other hand, an oral suspension properly compounded and well-agitated before use helps to avoid inhomogeneity. Nevertheless, this does not explain the wide difference observed between capsules and tablets even though the final step of manipulation is identical. This could be explained by differences in galenic formulations. Indeed, commercial tablets contain excipients (sodium laurylsulfate, hydrophobic colloidal silica) which could improve the recovery of APIs mixed with water by avoiding formation of aggregates or precipitates [33]. On the other hand, formulations of compounded capsules comprise only APIs and lactose as a diluent. There is no excipient which could improve recovery of APIs after opening the capsule. Many excipients could be used to improve the solubility of poorly soluble APIs in compounded capsules, such as colloidal silicon dioxide, hydroxyethylmethyl cellulose or methylcellulose. Nevertheless, further experiments are required to state if the use of such excipients could provide sufficient APIs dissolution to obtain accurate doses from compounded capsules. Additionally, milk could be used to reconstitute powder from capsules, but such a technique must be assessed first. Ours results are in accordance with Tuleu et al. who demonstrated that the extemporaneous preparation of nifedipine solution from crushed tablets leads to insufficient reproducibility and potential dosage errors [34]. Furthermore, this also leads to the modification of the dissolution profile provided by the original dosage form. Several studies have also shown that halving or quartering tablets was unreliable with a large deviation of weight. Watson et al. demonstrated that two different manipulations of 10 mg hydrocortisone tablets did not provide similar dose accuracy with large dose deviations associated with discrepancies between manipulations due to the dosage form, the prescribed dose or the experience of the manipulator [11]. In our study, all manipulations were performed by the same operator so as to avoid discrepancies due to manipulator skills and to focus only on administration techniques and dosage forms. Other studies, using different APIs or different fractioning methods, found similar results [12,13].

Here, we demonstrate that oral suspensions are a safer alternative to administering medicine when manipulations are required. Indeed, manipulations of capsules and tablets have been associated with underdosing and large variations in doses leading to the inadequate management of pediatric diseases. Nevertheless, use of oral suspensions could present several limits. First, stability studies are required to ensure that the preparation remains stable over time and to prevent underdosing or the appearance of toxic degradation products which could harm patients. Moreover, analytical control must be performed to assess the homogeneity of these preparations. As they are compounded with aqueous excipients, oral suspensions may be subject to microbiological growth. This microbiological contamination could occur during compounding, a risk that underscores the importance of using an adequate standard operating procedure to prevent it and of performing assays to control the microbiological quality of compounded oral suspensions [35]. Additionally, microbiological contaminations may occur during administration when administration devices are contaminated by the oral bacterial flora. In addition to physicochemical stability study and microbiological assays, real-life stability studies are needed to ensure that compounded oral suspensions are not at risk of oral microbiological contaminations when administered to patients. Furthermore, swallowing of oral suspensions could not be complete when administered as children, especially young ones, may be subject to choking or dripping out of the mouth. This aspect was not evaluated in this study, but it should be looked at to avoid underdosing.

## 5. Conclusions

Our study demonstrates that, when properly compounded, oral suspensions present higher benefits than manipulations of capsules and tablets, in terms of accuracy of the administered dose. This solution should be preferred when age-appropriate commercial medicines are not available.

Furthermore, this study also emphasizes the need to focus on techniques of administration when compounded dosage forms are required. More specifically, pharmacists must look beyond quality controls to extend their vision to “clinical pharmaceutical technology”.

## Figures and Tables

**Figure 1 pharmaceutics-13-01014-f001:**
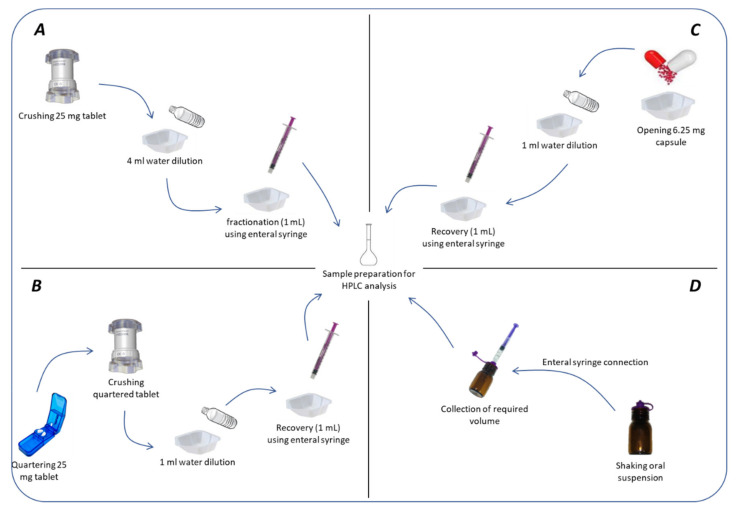
Techniques of administration for (**A**) crushed tablets; (**B**) quartered tablets; (**C**) compounded capsules; (**D**) compounded oral suspensions.

**Figure 2 pharmaceutics-13-01014-f002:**
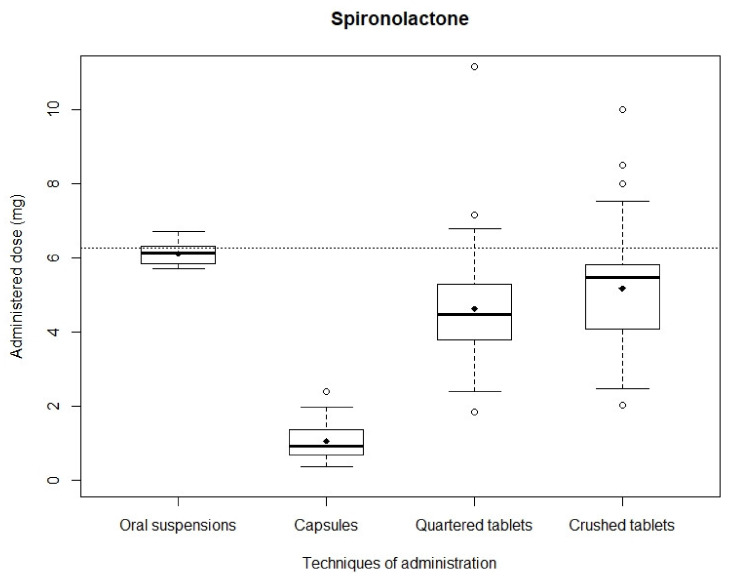
Distribution of spironolactone administered doses.

**Figure 3 pharmaceutics-13-01014-f003:**
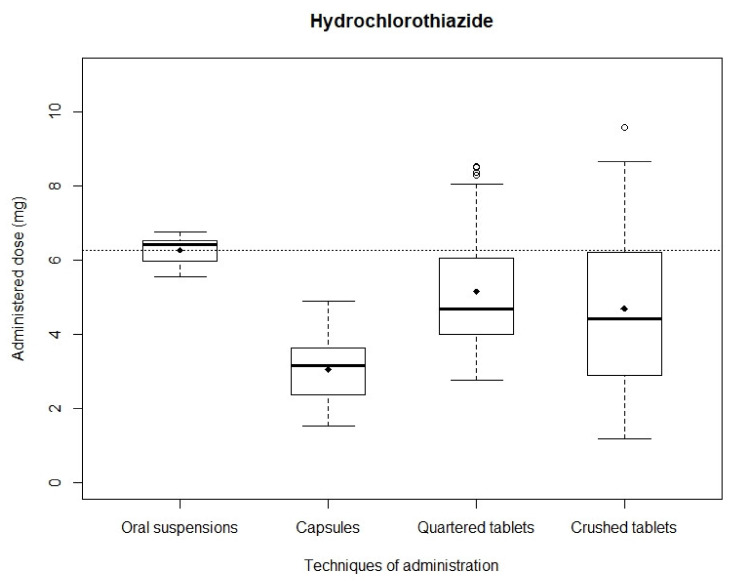
Distribution of hydrochlorothiazide administered doses.

**Table 1 pharmaceutics-13-01014-t001:** Average APIs content of capsules (mg, mean ± SD).

	Batch 1	Batch 2	Batch 3	Mean
Hydrochlorothiazide	5.91 ± 0.51	5.85 ± 0.57	6.11 ± 0.28	5.93 ± 0.44
Spironolactone	6.35 ± 0.45	6.12 ± 0.51	5.75 ± 0.46	6.07 ± 0.47

**Table 2 pharmaceutics-13-01014-t002:** Average APIs content of oral suspensions (mg/mL, mean ± SD).

	Batch 1	Batch 2	Batch 3	Mean
Hydrochlorothiazide	2.02 ± 0.04	1.96 ± 0,03	1.97 ± 0.08	1.98 ± 0.06
Spironolactone	5.10 ± 0.04	5.35 ± 0.13	4.74 ± 0.13	5.06 ± 0.28

## Data Availability

Not applicable.

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
