# Peer review of "Accuracy of Dose Administered to Children Using Off-Labelled or Unlicensed Oral Dosage Forms"

_pharmaceutics, 2021, doi:10.3390/pharmaceutics13071014_

Round 1
Reviewer 1 Report
This article compares the accuracy of administered dose provided by manipulation of two different dosage forms and compounded capsules and oral suspension in order to evaluate the practice of administering off-labelled mediacation in paedriatrics. This is an interesting topic to be studied as constitute an important problem for paediatric population.
I provide some recommendations:
- Keywords should include "off-labelled dosage forms"
- Introduction
- I suggest to make reference to the 3D printing of medicines as a tool for personalized medicine in paediatrics.
- Line 42: please delete "at"
- Line 68: please delete "of"
- Line 72: it is not clear the following part of the aim of the study "...propose an alternative to reduce dose issues associated with the use...
- Materials and Methods
- In the first paragraph. Could authors explain better why they have decided to employ 6.25 mg of dose? What is the range of doses required for neonates?
- Please delete from line 178 to line 185
- Conclusions
- The claim that there is the first study comparing dose accuracy obtained from manipulations of different dosage forms should be softened since for example there is an an article that compares four different types of aspirin tablets (two dispersible, one conventional and one chewing) (Journal of Paediatrics109(11), pp. 2394-2401)
- The claim that there is the first study comparing dose accuracy obtained from manipulations of different dosage forms should be softened since for example there is an an article that compares four different types of aspirin tablets (two dispersible, one conventional and one chewing) (Journal of Paediatrics109(11), pp. 2394-2401)
Author Response
- This article compares the accuracy of administered dose provided by manipulation of two different dosage forms and compounded capsules and oral suspension in order to evaluate the practice of administering off-labelled mediacation in paedriatrics. This is an interesting topic to be studied as constitute an important problem for paediatric population.
Thank you very much for your valuable and constructive suggestions. We agree with most of your comments, and we have carefully revised the manuscript according to your suggestions.
I provide some recommendations:
- Keywords should include "off-labelled dosage forms"
We have added this keyword to the manuscript.
- Introduction
- I suggest to make reference to the 3D printing of medicines as a tool for personalized medicine in paediatrics.
We have added reference on 3D printing as suggested (line 368-369)
- Line 42: please delete "at"
We have deleted « at » at line 42.
- Line 68: please delete "of"
We have deleted « of » at line 68.
- Line 72: it is not clear the following part of the aim of the study "...propose an alternative to reduce dose issues associated with the use...
We agree with your comment. We have clarified this sentence in the manuscript (line 72 to 74).
- Materials and Methods
- In the first paragraph. Could authors explain better why they have decided to employ 6.25 mg of dose? What is the range of doses required for neonates?
As suggested, we have added the range of doses used for neonates in the manuscript and we have explained the choice of 6.25 mg (lines 83 to 86).
- Please delete from line 178 to line 185
We have deleted these lines.
- Conclusions
- The claim that there is the first study comparing dose accuracy obtained from manipulations of different dosage forms should be softened since for example there is an an article that compares four different types of aspirin tablets (two dispersible, one conventional and one chewing) (Journal of Paediatrics109(11), pp. 2394-2401)
We agree with this suggestion. We have softened our conclusion by removing this claim.

Reviewer 2 Report
In the submitted manuscript authors have addressed the accuracy of the dose administered to children using off-labeled or unlicensed oral dosage forms. This is a relevant topic worthy of investigation. I only have some minor suggestions for authors.
Firstly, I would recommend authors to include some additional data (references) on the dosing of the two selected APIs.
It is not clear what is meant by the "squared" tablets that authors refer to.
In lines 178-185 (pages 4-5) instructions for the manuscript preparation should be deleted.
Could authors suggest some other excipients (in addition to the lactose) that could be used to formulate oral capsules in order to improve the solubility of poorly soluble APIs, such was the case in the presented study? And also, could another type of vehicle be used to dilute, i.e. reconstitute powder from capsules that could provide sufficient API dispersion (similar to compounded suspension)? This way benefits of the solid dosage form and suspensions could be exploited.
Author Response
- In the submitted manuscript authors have addressed the accuracy of the dose administered to children using off-labeled or unlicensed oral dosage forms. This is a relevant topic worthy of investigation. I only have some minor suggestions for authors.
Thank you very much for your valuable and constructive suggestions. We agree with your comments, and we have carefully revised the manuscript according to the suggestions.
- Firstly, I would recommend authors to include some additional data (references) on the dosing of the two selected APIs.
References on the dosing of the two selected APIs is already presented in the manuscript (reference 18, Segar JL. Neonatal Diuretic Therapy: Furosemide, Thiazides, and Spironolactone. Clin Perinatol 2012;39:209–20.).
Nevertheless, we have added some additional data in the manuscript (lines 83 to 86 + van der Vorst, M.M.J.; Kist, J.E.; van der Heijden, A.J.; Burggraaf, J. Diuretics in Pediatrics. Pediatr. Drugs 2006, 8, 245–264, doi:10.2165/00148581-200608040-00004.)
- It is not clear what is meant by the "squared" tablets that authors refer to.
We used the wrong word, we wanted to say “scored tablets”. We have modified this part in the manuscript (line 82)
- In lines 178-185 (pages 4-5) instructions for the manuscript preparation should be deleted.
We have deleted these lines.
- Could authors suggest some other excipients (in addition to the lactose) that could be used to formulate oral capsules in order to improve the solubility of poorly soluble APIs, such was the case in the presented study? And also, could another type of vehicle be used to dilute, i.e. reconstitute powder from capsules that could provide sufficient API dispersion (similar to compounded suspension)? This way benefits of the solid dosage form and suspensions could be exploited.
Thank you for your suggestions. We have suggested some excipients and vehicles which could improve the dissolution of APIs from compounded capsules in the manuscript (lines 326 to 331)

Round 2
Reviewer 1 Report
Authors have adopted the majority of the recommendations suggested in the review report. However, there are two important issues that in my opinion should be addressed. These issues affect the introduction, the aim of the study and the conclusions so I think that they must be addressed before publication. These recommendations are:
- Introduction: I suggested to make reference to the 3D printing of medicines as a tool for personalized medicine in paediatrics. However the reference to the 3D printing of medicines has been made in the conclusions.
- Aim of the study: I advised authors that it was not clear the following part of the aim of the study "...propose an alternative to reduce dose issues associated with the use..." However, they repeat the same expresion that is in bold letter and that in my opinion is not clear. For example it could have changed to "... propose an alternative to reduce dose inaccuracy associated with the use..."
Author Response
Authors have adopted the majority of the recommendations suggested in the review report. However, there are two important issues that in my opinion should be addressed. These issues affect the introduction, the aim of the study and the conclusions so I think that they must be addressed before publication. These recommendations are:
Thank you very much for these additional suggestions. We agree with your comments and have revised the manuscript according to your suggestions.
- Introduction: I suggested to make reference to the 3D printing of medicines as a tool for personalized medicine in paediatrics. However the reference to the 3D printing of medicines has been made in the conclusions.
We have moved the reference to 3D-printing from the discussion to the introduction (lines 56 to 58).
- Aim of the study: I advised authors that it was not clear the following part of the aim of the study "...propose an alternative to reduce dose issues associated with the use..." However, they repeat the same expresion that is in bold letter and that in my opinion is not clear. For example it could have changed to "... propose an alternative to reduce dose inaccuracy associated with the use..."
We have clarified the aim of the study, following your constructive suggestions (line 75).
